# Understanding and Leveraging Expert-level Monosemanticity in Finetuning MoE LLMs

## Abstract

Large language models (LLMs) with Mixture-of-Experts (MoE) architectures have emerged as a promising approach for enhancing scalability and efficiency, with minimal performance degradation across diverse downstream tasks. However, the interpretability of experts and efficient post-training methods of domain experts remain understudied. In this paper, we first analyze the expert-level monosemanticity of MoE based on the sparse autoencoder (SAE), thereby facilitating a deeper understanding of domain experts' roles. Additionally, leveraging the enhanced monosemanticity induced by the sparse activations of MoE LLMs, we propose a new fine-tuning strategy that freezes domain-agnostic experts in specific layers. Unlike dense LLMs, the sparsity of MoE enables experts to exhibit stronger expert-level monosemantic behavior, allowing us to identify experts responsible for particular downstream tasks and freeze those unrelated during post-training. By only updating domain-relevant experts, our method mitigates the risk of catastrophic forgetting in other domains and reduces computational costs. Empirically, we apply this strategy to supervised fine-tuning of MoE models on medicine and tool-use data. Results show that monosemanticity-guided tuning achieves performance comparable to fully-tuned models on medicine tasks, while preserving better performance in other domains(21.19%). Our study provides an interpretability-guided strategy for understanding and finetuning MoE LLMs while alleviating performance degradation across domains.

## 1 Introduction

Mixture-of-Experts (MoE) architectures have recently emerged as a promising paradigm for scaling foundation models such as Kimi-K2 (Team et al., 2025), DeepSeek-V3 (Liu et al., 2024), and Qwen3 (Yang et al., 2025). By activating only a subset of experts in each layer, MoE substantially reduces computational overhead while preserving the overall capacity of the model. This sparsity allows training models with hundreds of billions of parameters without a proportional increase in inference cost, establishing MoE as a central design choice in state-of-the-art large-scale language models. Beyond efficiency, MoE offers a natural mechanism for expert specialization across different domains or tasks, which holds promise for improving the interpretability of these models (Jafar et al., 2025). Nevertheless, systematic expLoRAtions into how to better understand and leverage this interpretability remain limited.

A key property in analyzing model interpretability is monosemanticity, where each dimension is activated primarily by a single natural concept (Elhage et al., 2022; Cunningham et al., 2023), such as sentiment polarity, semantic categories, or word senses. The conventional approach to enhancing monosemanticity is to train a sparse autoencoder (SAE) on the internal activations of models. However, SAEs are post-hoc methods that introduce additional computational overhead. Prior work has shown that sparsity in the SAE feature space is closely linked to improved monosemanticity (Gao et al., 2024; Cunningham et al., 2023). This motivates us to ask whether the inherent sparsity in MoE language models similarly promotes monosemantic representations. In this paper, we propose a new evaluation framework to evaluate the monosemanticity of MoE models in specific downstream domains. Our findings reveal that increased sparsity significantly strengthens expert-level monosemanticity and drives experts to specialize across different downstream domains.

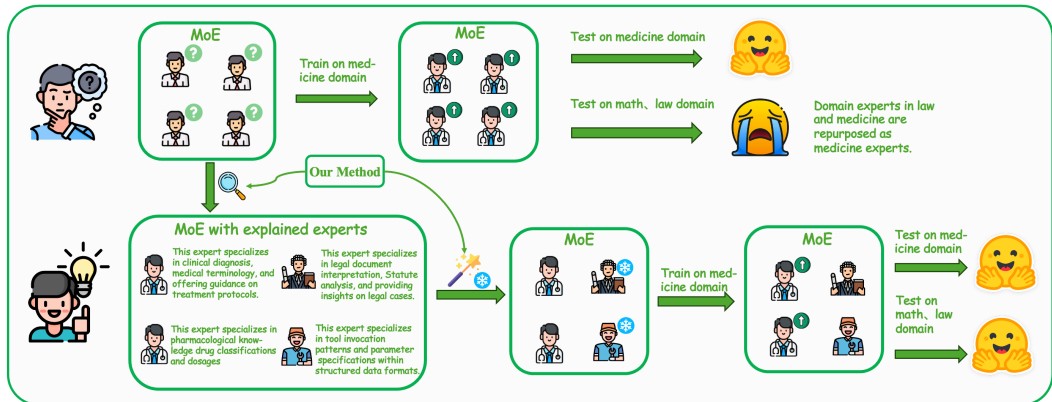

Figure 1: **(Top)** Training without our method: Expert functions remain uncharacterized, causing all experts to be fine-tuned on the target domain. This disrupts their original specializations, leading to degraded performance in other domains. **(Bottom)** Training with our method: We first generate explanations to reveal expert specializations, then leverage fine-grained metrics to selectively fine-tune relevant experts while freezing others. This preserves expert specialization and maintains performance across domains.

Building on our analysis of enhanced monosemanticity in MoE models, we further examine how this property can inspire better post-training strategies for large language models. A well-known challenge in post-training, particularly supervised fine-tuning, is that optimizing for a specific downstream task often comes at the expense of performance in other domains (Dong et al., 2023; Kumar et al., 2022). This raises the question of how to preserve cross-domain generalization while still achieving strong task-specific improvements. Monosemanticity provides a natural lens to address this issue: with MoE architectures, the increased sparsity encourages experts to specialize in distinct domains, suggesting that only a subset of experts is truly responsible for a given downstream task.

As shown in Figure 1, motivated by this observation, we propose a selective post-training strategy in which only the experts relevant to the target task are fine-tuned, while the others remain frozen. To identify these task-relevant experts, we combine monosemanticity scores with activation frequencies, allowing us to isolate experts that are both highly active in the target domain and minimally involved in irrelevant tasks. Empirically, we apply this strategy to supervised fine-tuning of MoE models on tool-use datasets. Our results demonstrate that monosemanticity-guided tuning achieves performance comparable to fully fine-tuned models on tool-use tasks, while better preserving performance in other domains. By integrating monosemantic interpretability into post-training design, our approach offers a principled means of improving task-specific performance while mitigating negative transfer across domains. Our contributions can be summarized as follows:

- We analyze monosemanticity in MoE LLMs and introduce a novel evaluation framework, showing that sparsity at the expert level significantly enhances monosemanticity. In addition, our framework provides insights into how monosemanticity varies across different layers of the model, how it differs among experts, and how these patterns relate to downstream task specialization.

- We develop a metric to identify task-relevant monosemantic experts by combining monosemanticity scores with activation frequencies. Based on that, we propose a new post-training strategies of MoE LLMs that only optimizes the experts related to targeted downstream tasks.

- We empirically validate our approach on supervised fine-tuning, and results demonstrate that our strategy achieves comparable performance on target tasks as fully fine-tuning while better preserving performance in other domains, such math, computer science, and history.

## 2 RELATED WORK & PRELIMINARY

### 2.1 MOE LLMS

Sparse MoE LLMs have achieved significant scalability by activating only a subset of parameters in each layer through a routing network (Team et al., 2025; Liu et al., 2024). This modular design enables efficient scaling and introduces additional flexibility for model adaptation. Formally, in the $l$-th layer of the MoE, the output hidden state of the $t$-th token, $h_t^l$, is computed as:

$$h_t^l = \sum_{i=1}^{N} g_{i,t} \cdot \text{FFN}_i(u_t^l) + u_t^l \tag{1}$$

Here, $N$ denotes the total number of experts, $\text{FFN}_i(\cdot)$ represents the $i$-th expert network, and $g_{i,t}$ is the gating value for the $i$-th expert when processing the $t$-th token. The gating mechanism is defined as:

$$g_{i,t} = \begin{cases} s_{i,t}, & \text{if } s_{i,t} \in \text{TopK}(\{s_{j,t}|1 \le j \le N\}, K) \\ 0, & \text{otherwise} \end{cases} \tag{2}$$

where $s_{i,t}$ denotes the token-expert gating score that determines the routing of token $t$ to expert $i$, and the function $\text{TopK}(\cdot, K)$ selects the $K$ highest gating scores from the complete set of expert scores, effectively determining which experts will be activated for processing the current token.

### 2.2 FINETUNING OF MOE LLMS

Fine-tuning strategies for MoE LLMs are widely regarded as more complex than those for dense LLMs (Cai et al., 2025). Prior research has investigated expert-specific tuning (Bai et al., 2025; Wang et al., 2024), dynamic data mixing (Zhu et al., 2024), and router adaptation (Liu et al., 2025). A common challenge in these approaches is identifying which experts are most relevant to a target domain. To this end, existing methods rely on black-box statistical metrics—such as expert activation frequency or router output probabilities (Cai et al., 2025)—to select domain-specific experts. However, such metrics provide little interpretability and often fail to disentangle domain-specific from shared experts. Consequently, the interpretability of individual experts in MoE LLMs remains underexplored, limiting their broader applicability. In contrast, in this paper, we leverage the principle of interpretability inspired by sparse autoencoders (Cunningham et al., 2023) to design interpretable expert-level fine-tuning strategies for MoE LLMs.

### 2.3 MONOSEMANTICITY IN LARGE LANGUAGE MODELS

Although large language models (LLMs) have achieved remarkable performance across diverse domains (Liu et al., 2024; Yang et al., 2025), their decision-making processes remain opaque. To shed light on this black box, recent studies have introduced SAEs (Gao et al., 2024; Cunningham et al., 2023), which induce monosemantic dimensions where each unit tends to capture a specific natural concept (Shu et al., 2025). Formally, consider a hidden activation vector $z \in \mathbb{R}^d$ inside the model. A SAE maps this vector through an encoder–decoder pipeline: the encoder transforms $z$ into a sparse latent embedding, and the decoder then attempts to reconstruct the original representation from that embedding. As a concrete example, in a top-K SAE (Gao et al., 2024), this process can be written as:

$$\begin{aligned} h &= \text{TopK}(W_{enc}z - b), \\ \hat{z} &= W_{dec}h + b. \end{aligned} \tag{3}$$

To obtain the encoded representation $h$, the SAE applies a linear map parameterized by $W_{enc} \in \mathbb{R}^{s \times d}$ together with a bias term $b \in \mathbb{R}^s$. The decoder then maps this sparse embedding back into the original feature space through $W_{dec} \in \mathbb{R}^{d \times s}$. Training proceeds by optimizing the reconstruction objective:

$$\mathcal{L}_{\text{SAE}}(W_{enc}, W_{dec}, b) = \|\hat{z} - z\|^2.$$

SAEs have demonstrated effectiveness in interpretability-related applications such as task-oriented data selection (Ma et al., 2025), vision–language disentanglement (Pach et al., 2025), latent semantic decomposition (Lan et al., 2024), and privacy analysis of LLMs (Frikha et al., 2025). Furthermore, the monosemanticity revealed by SAEs has been leveraged to construct interpretable reward models

that provide transparent, concept-based feedback (Zhang et al., 2025; Li et al., 2025). Collectively, these works highlight the feasibility of applying SAE-based interpretability in LLMs. However, most existing studies focus exclusively on dense LLMs, overlooking the expert-level structure of MoE architectures. Consequently, in this paper, we focus on the expert-level monosemanticity in MoE LLMs and leverage it to design interpretable fine-tuning strategies, enabling more principled expert selection and adaptation.

## 3 EXPERT-LEVEL MONOSEMANTICITY IN MOE LLMS

SAEs have recently emerged as powerful tools for improving the interpretability of LLMs (Cunningham et al., 2023). SAEs encourage monosemanticity by enforcing sparsity, such that each hidden dimension tends to represent a distinct natural concept (Gao et al., 2024). Notably, MoE LLMs inherently exhibit sparsity through their expert selection mechanism, activating only a subset of experts for each input. This parallel motivates our central research question: Does the inherent sparsity in MoEs lead to stronger monosemanticity? To address this, Section 3.1 introduces a systematic framework for evaluating monosemanticity in MoE LLMs, while Section 3.2 provides empirical insights into how monosemanticity emerges within MoE architectures.

### 3.1 EVALUATION FRAMEWORK FOR MONOSEMANTICITY IN MOE LLMS

To investigate monosemanticity in MoE LLMs, we propose an evaluation framework grounded in structural analogies between SAE and MoE architectures. Specifically, we map the sparse gating outputs in MoEs to the sparse intermediate representations in SAEs; the top-k activation mechanism in MoEs mirrors the thresholded activation in SAEs; and each MoE expert is considered analogous to a single SAE neuron. Building on these correspondences, we extend the notion of monosemanticity from the neuron level in SAEs to the expert level in MoE models. Inspired by automated SAE monosemanticity evaluation methods (Paulo et al., 2025), we present a systematic, expert-level evaluation framework for MoE LLMs.

The proposed framework consists of two stages. First, we employ a three-stage sampling strategy to construct expert activation datasets, including top-activated samples, importance-weighted samples, and randomly selected negatives. Natural language explanations for each expert are then generated using two complementary prompts: $P_{mono}$, providing a general characterization, and $P_{domain}^{(d)}$, highlighting domain-specific semantics. In the second stage, an external LLM predicts expert activation based on these explanations, formulated as a binary classification task whose accuracy reflects the degree of expert-level monosemanticity. Complete prompt templates and design details are provided in Appendix A.1, with detailed expert explanations in Appendix A.3.

To quantitatively assess the monosemanticity of MoE, we propose two key metrics: the expert monosemanticity score (EMS) and the expert domain monosemanticity score (EDMS). EMS measures the degree to which an expert exhibits monosemanticity across a general domain, while EDMS evaluates monosemanticity within a specific downstream domain. Formally, the EMS for expert $E_i$ is defined as:

$$\text{EMS}_i = \frac{1}{|\text{E}_i|} \sum_{s_j \in \mathcal{D}} \mathbb{I}[\hat{y}_{i,j}^{mono} = y_{i,j}], \qquad (4)$$

where $\mathcal{D}$ denotes the test dataset and $j$ indexes its samples. Here, $\hat{y}_{i,j}^{mono}$ represents the binary prediction of whether expert $E_i$ would be activated for sample $s_j$, based on the explanation derived from prompt $P_{mono}$, and $y_{i,j}$ is the corresponding ground-truth activation label. Higher EMS values indicate stronger monosemanticity.

For evaluating monosemanticity in a specific domain $d$, the EDMS is defined as:

$$\text{EDMS}_i^{(d)} = \frac{1}{|\text{E}_i|} \sum_{s_j \in \mathcal{D}} \mathbb{I}[\hat{y}_{i,j}^{(d)} = y_{i,j}^{(d)}]. \qquad (5)$$

Here, $\hat{y}_{i,j}^{(d)}$ denotes the prediction of expert $E_i$ for domain $d$, and $y_{i,j}^{(d)}$ is the corresponding ground-truth label. A high EDMS indicates that the expert is monosemantic with respect to specific concepts in the target domain.

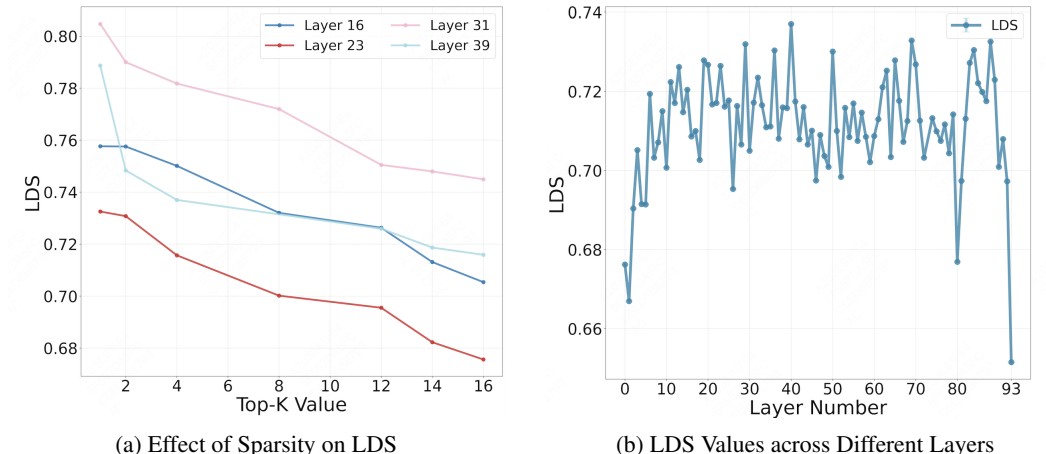

(a) Effect of Sparsity on LDS

(b) LDS Values across Different Layers

Figure 2: Analysis of sparsity and monosemanticity in MoE experts. The metric layer-wise decoupling score (LDS) describes the average monosemanticity of a layer. Panel (a) examines the sparsity-monosemanticity relationship, showing that higher sparsity enhances expert specialization. Panel (b) tracks monosemanticity variation across network depth, revealing that middle layers exhibit optimal monosemanticity.

## 3.2 EMPIRICAL INSIGHTS OF EXPERT-LEVEL MONOSEMANTICITY

To advance the understanding of expert monosemanticity and sparsity in MoE models, and to lay the groundwork for further applications, we systematically analyze and uncover three key empirical insights in this section using the proposed evaluation framework.

> **Insight 1:** *The sparse architecture of MoE LLMs not only enhances computational efficiency but also induces a higher degree of expert monosemanticity, distinguishing them from dense LLMs.*

**Clarification for Insight 1:** Theoretical and empirical findings from SAE research establish that higher sparsity promotes stronger monosemanticity (Gao et al., 2024). In MoE LLMs, this sparsity is intrinsically determined by the number of activated experts, where a greater number of activations corresponds to lower sparsity. Building on this principle, our work investigates the relationship between expert-level sparsity and monosemanticity through controlled experiments on the Qwen3-30B-A3B model (Yang et al., 2025). We select four representative layers and systematically vary the number of activated experts in each. Monosemanticity is quantified using the Layer-wise Decoupling Score (LDS), which we define as the average EMS of all experts within a layer:

$$\text{LDS}_l = \frac{1}{N_l} \sum_{i=1}^{N_l} \text{EMS}_i^{(l)}, \tag{6}$$

where $\text{LDS}_l$ denotes the LDS for the $l$-th layer and $N_l$ is the number of experts in that layer. LDS captures the average monosemanticity of experts at a given layer. As shown in Figure 2a, increasing the number of activated experts (i.e., reducing sparsity) consistently and significantly decreases LDS across all layers, indicating progressively weaker functional separation and diminished expert-level monosemanticity. Building directly on this key observation, we next analyze how monosemanticity varies across different layers of the MoE architecture.

> **Insight 2:** *Middle layers in MoE LLMs exhibit the highest degree of monosemanticity, while input and output layers show nearly the lowest. This suggests that experts in the middle layers are most decoupled and specialized, making them especially well-suited for operations that leverage monosemantic representations.*

**Clarification for Insight 2:** Figure 2b presents the LDS scores across network depth for Qwen3-235B-A22B. We observe a rapid increase in LDS in the early layers, a consistently high plateau across the middle layers, and a sharp decline near the output. This trajectory suggests a three-stage organization: early layers focus on disentangling basic features, middle layers achieve the strongest

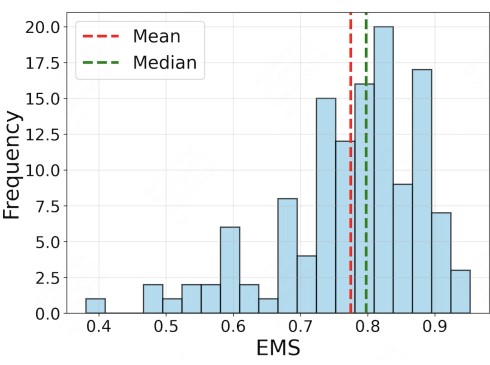 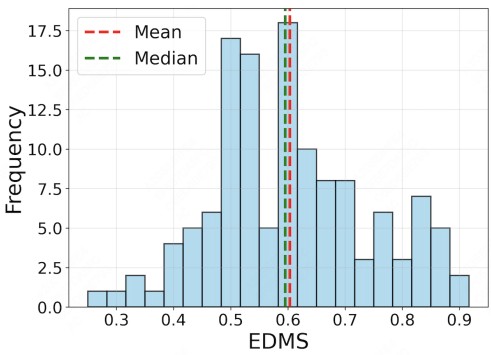

(a) Expert monosemanticity scores        (b) Expert domain monosemanticity scores

Figure 3: Distribution of expert monosemanticity in the middle layer. Panel (a) evaluates the distribution of general expert monosemanticity scores (EMS) across all domains. Panel (b) evaluates the distribution of domain-specific expert domain monosemanticity scores (EDMS) for the tool-use domain.

expert decoupling and monosemanticity, and output layers gradually lose this separation. These findings highlight the central role of middle-layer experts in maintaining semantic specialization, i.e., monosemanticity, within the model. Besides patterns of monosemanticity across depth, we next examine domain-level monosemanticity to explore how expert specialization varies under different types of input data. To verify the stability of the metrics when using cheaper external LLMs, we added experiments in Appendix A.5.

> **Insight 3:** *The degree of monosemanticity among MoE experts varies substantially across domains. Some experts exhibit consistently strong monosemanticity across all settings, while others are highly specialized within particular domains.*

**Clarification for Insight 3:** To investigate domain-dependent variations in expert monosemanticity, we apply our monosemanticity evaluation framework using both general prompts ($P_{mono}$) and tool-use-specific prompts ($P_{domain}^{(d)}$) to the middle layer of Qwen3-MoE-30B-A3B, computing EMS and EDMS for all 128 experts. As shown in Figure 3a and Figure 3b, EDMS scores are generally lower than EMS scores, suggesting that while some experts maintain high monosemanticity, their activity often pertains to domains outside of tool use. We also identify polysemantic experts who are activated across multiple domains. Overall, these findings reveal the coexistence of domain-specialized and shared experts, indicating that experts exhibit distinct patterns of activation across different domains.

## 4 MONOSEMANTICITY-GUIDED FINE-TUNING STRATEGY FOR MOE LLMS

During the fine-tuning process of MoE LLMs, it is widely observed that tuning on a target task will hurt the performance in other downstream domains (Dong et al., 2023). Intuitively, as we have observed that there exist monosemantic experts, we can freeze the irrelevant experts and only update the experts related to the tool-use to mitigate this phenomenon. In Section 4.1, we present our method for identifying monosemantic experts, and in Section 4.2, we provide a theoretical justification for this approach.

### 4.1 EXPERT SELECTION STRATEGY BASED ON MONOSEMANTICITY SCORES

In previous works (Bai et al., 2025; Wang et al., 2024), the commonly used method is calculating the activation frequency (AF) to find important experts:

$$\text{AF}_i^{(d)} = \frac{1}{|T_d|} \sum_{t \in T_d} \mathbb{I}[E_i'(t)], \tag{7}$$

where $E_i'$ indicates whether expert $E_i$ is activated for token $t$, $T_d$ denotes the set of tokens in the fine-tuned domain, and $\mathbb{I}[\cdot]$ is the indicator function. We know that there exist polysemantic experts who are activated across multiple semantics. Fine-tuning such experts will influence performance in other domains. Therefore, it is crucial to distinguish between monosemantic and polysemantic experts with high activation frequency.

Leveraging our evaluation framework and the domain-specific prompt $\mathrm{P}_{domain}^{(d)}$, we can identify monosemantic experts that encode semantic features relevant to the fine-tuned domain (e.g., medicine or tool-use). These experts are key to domain adaptation and are less likely to interfere with other domains. Building on this, we introduce the expert domain monosemantic responsibility score (EDMRS) and its associated strategy. EDMRS combines an expert's activation frequency (AF) with its EDMS, weighting the importance of each expert's activations. Formally, the EDMRS for expert $E_i$ in domain $d$ is defined as:

$$\mathrm{EDMRS}_i^{(d)} = \mathrm{AF}_i^{(d)} \times \mathrm{EDMS}_i^{(d)}$$
$$= \frac{\mathrm{EDMS}_i^{(d)}}{|T_d|} \sum_{t \in T_d} \mathbb{I}[E_i'(t)]$$

where $\mathrm{EDMS}_i^{(d)}$ serves as a monosemanticity-based weighting coefficient, thereby increasing the importance of monosemanticity. A high EDMRS score indicates that the expert predominantly processes key semantic tokens relevant to the fine-tuned domain and remains functionally decoupled from other domains.

Benefiting from the structural properties of MoE expert networks and our design, our method can be orthogonally combined with fine-tuning approaches such as LoRA rather than being mutually exclusive. Since we essentially select parameters that are relevant to the target domain and decoupled from other domains, these fine-tuning methods can treat the selected parameters as a new sub-network; for example, LoRA's low-rank matrices can be directly applied to them.

In summary, the EDMRS strategy refines the traditional AF-based expert selection by explicitly incorporating both domain specificity and functional independence into the selection criterion. By jointly considering an expert's activation frequency and its domain monosemanticity, EDMRS ensures that the selected experts are not only highly relevant to the target domain but also minimally entangled with other domains. This targeted selection facilitates effective domain adaptation and fine-tuning, leading to enhanced performance in the fine-tuned domain while substantially reducing cross-domain interference.

## 4.2 THEORETICAL VERIFICATION OF EDMRS STRATEGY

As discussed earlier, we propose a new strategy to identify monosemantic experts related to a certain downstream domain. In this section, we provide a theoretical justification for our strategy.

Specifically, we denote $d$ as the fine-tuned target domain, and consider a layer of the MoE model containing the expert set $E_1, E_2, \ldots, E_N$. For each expert $E_i$, we define its activation frequency in domain $d$ as $\mathrm{AF}_i^{(d)}$ and its Expert Domain Monosemanticity Score as $\mathrm{EDMS}_i^{(d)}$. To formalize the objective for expert selection, we define the performance change resulting from fine-tuning expert $E_i$ in domain $d$ for a single input token $t$ as:

$$C_i(t) = \Delta_{\mathcal{P}}^{(d)}(E_i, t) - \Delta_{\mathcal{N}}^{(\mathrm{other})}(E_i, t) \tag{8}$$

where $\Delta_{\mathcal{P}}^{(d)}(E_i, t)$ represents the performance improvement of $E_i$ on token $t$ in the fine-tuned domain $d$, while $\Delta_{\mathcal{N}}^{(\mathrm{other})}(E_i, t)$ denotes the performance degradation of $E_i$ on token $t$ in other domains.

Within our MoE monosemanticity valuation framework, a high $\mathrm{EDMS}_i^{(d)}$ score indicates that the token activation pattern of expert $E_i$ closely aligns with the domain-specific explanation. In other words, the token features processed by the expert are well-matched to the fine-tuned domain $d$, contributing substantially to domain-specific performance while remaining largely decoupled from other domains. Conversely, a low $\mathrm{EDMS}_i^{(d)}$ implies that the expert's activations do not correspond to the domain explanation, with token features poorly aligned to domain $d$, resulting in minimal domain-specific contribution and increased coupling with other domains.

Building upon this theoretical intuition, we therefore posit the following hypothesis: that the average **positive** change in performance for a single token, when conditioned on the activation of expert $E_i$, exhibits a positive correlation with the value of $\text{EDMS}_i^{(d)}$.

**Assumption 1.**

$$\mathbb{E}[\Delta_{\mathcal{P}}^{(d)}(E_i, t) \mid E_i'] = k_1 \cdot EDMS_i^{(d)} \tag{9}$$

$$\mathbb{E}[\Delta_{\mathcal{N}}^{(other)}(E_i, t) \mid E_i'] = k_2 \cdot (1 - EDMS_i^{(d)}) \tag{10}$$

*where $k_1, k_2 > 0$ are constants.*

We next present the following theorem that characterizes the performance of our strategy:

**Theorem 1** (EDMRS Optimal Expert Selection Theorem). *Under Assumption 1, given a constraint on the number of selected experts $m$, selecting the $m$ experts with the highest EDMRS scores $EDMRS_i^{(d)} = AF_i^{(d)} \cdot EDMS_i^{(d)}$ for fine-tuning maximizes the expected performance improvement in the fine-tuned domain $d$:*

$$S^* = \arg \max_{|S|=m} \sum_{i \in S} \mathbb{E}[C_i]$$
$$= \arg \max_{|S|=m} \sum_{i \in S} EDMRS_i^{(d)}$$

*where $S^*$ denotes the optimal expert selection set, and $\mathbb{E}[C_i] = \mathbb{E}[C_i(t)]$ is the expected total performance change of expert $E_i$ across all domains.*

This theorem demonstrates that our EDMRS-based expert selection strategy theoretically maximizes performance both in the fine-tuned domain and across other domains, providing a formal justification for our approach. A detailed proof is presented in Appendix A.4.

## 5 EXPERIMENTS

For SFT, we perform our experiments using the MedMCQA and the tool-use training dataset (Qian et al., 2025; Pal et al., 2022). For evaluation, the BFCL V3 single-turn dataset and the MedMCQA validation set serves as the in-domain test set, while subsets from MMLU-Pro (e.g., history, law) are used as out-of-domain test sets to measure potential performance degradation after fine-tuning (Patil et al., 2025). Additional details on the experimental setup are provided in Appendix A.6.

### 5.1 COMPARISONS ON MoE FINE-TUNING STRATEGIES

To evaluate the impact of post-training on performance in other domains, we perform expert-selection fine-tuning across all layers for the medicine domain, and for the tool-use domain, we select the layer with the highest LDS. We compare the effectiveness of our proposed EDMRS strategy against conventional approaches by considering three strategies: (1) Full-parameter Strategy: all model parameters are fine-tuned, serving as the baseline; (2) AF Strategy: in the selected intermediate layer, the 16 experts with the highest activation frequency are fine-tuned; (3) EDMRS Strategy: in the same layer, the 16 experts with the highest EDMRS scores are fine-tuned while all other experts remain frozen.

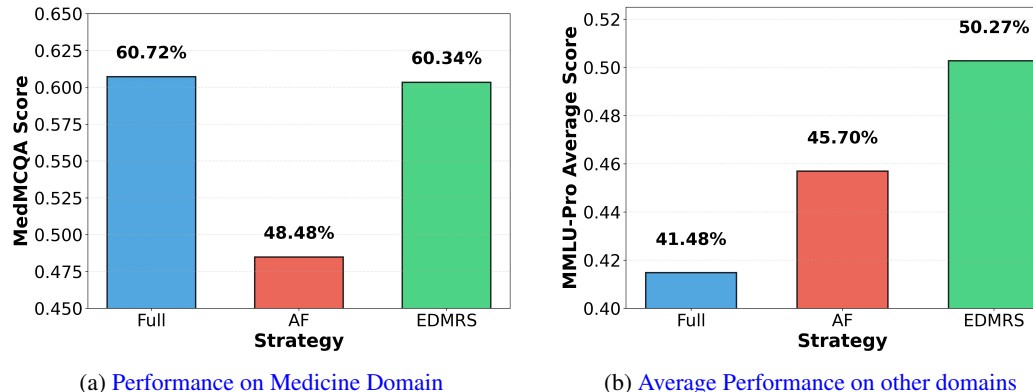

(a) Performance on Medicine Domain

(b) Average Performance on other domains

Figure 4: Comparative analysis of fine-tuning strategies in the medicine domain and related generalization. We compare three strategies: 1) a full-parameter fine-tuning strategy, 2) a strategy based on activation frequency, and 3) our strategy that identifies monosemantic experts with EDMRS scores. Panel (a) evaluates medicine domain performance across strategies, Panel (b) measures cross-domain generalization on MMLU-Pro subsets.

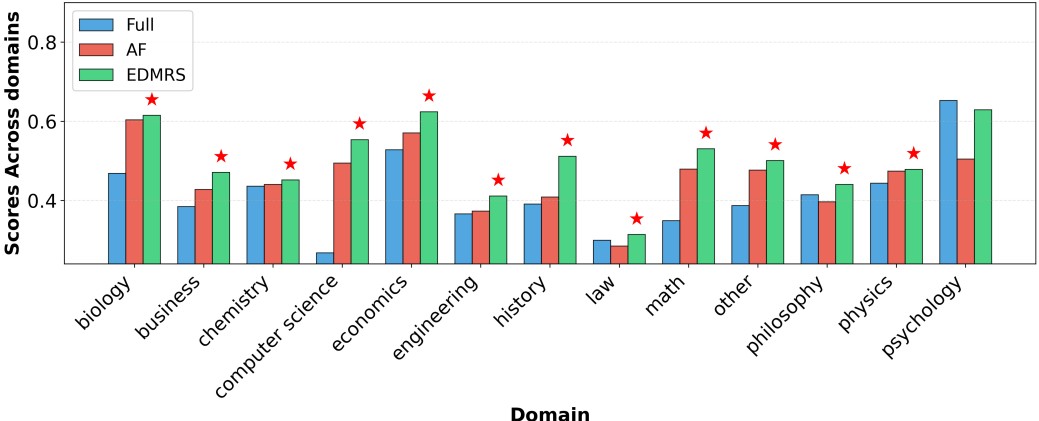

Figure 5: Comparative analysis of fine-tuning strategies on non-fine-tuned domains

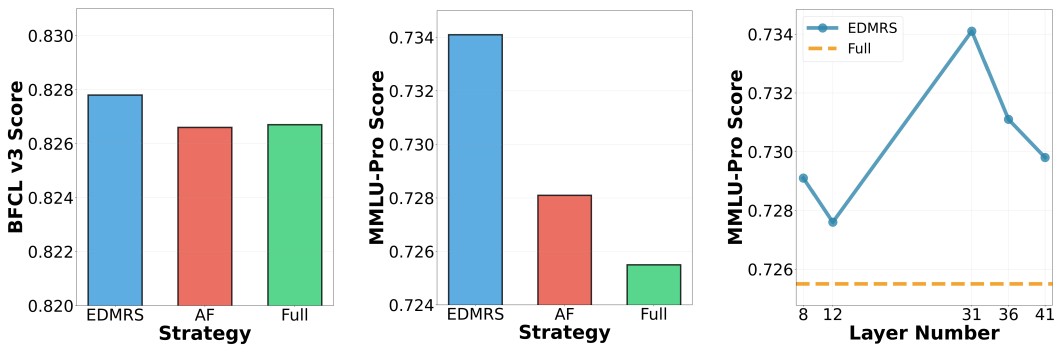

(a) Performance on Tool-use Domain

(b) Performance on other domains

(c) EDMRS strategy at different Layers

Figure 6: Comparative analysis of fine-tuning strategies in the tool-use domain and associated generalization across domains and layers. We compare three strategies: 1) a full-parameter fine-tuning strategy, 2) a strategy based on activation frequency, and 3) our strategy that identifies monosemantic experts with EDMRS scores. Panel (a) evaluates tool-use domain performance across strategies, Panel (b) measures cross-domain generalization on MMLU-Pro subsets, Panel (c) analyzes the layer-wise effectiveness of our proposed strategy.

Figure 4a compares the in-domain performance on the medicine domain. EDMRS (60.34%) substantially outperforms AF (48.48%) and approaches Full (60.72%). More importantly, EDMRS demonstrates superior generalization on the out-of-domain MMLU-Pro benchmark, achieving 50.27% and surpassing the Full-parameter (41.48%) and AF (45.70%) strategies by relative margins of 21.2% and 10.0%, respectively. Figure 5 shows the performance on each specific domain, where EDMRS is clearly the best. Notably, EDMRS outperforms the Full by 106.37% on computer science and by 51.89% on math, reflecting that EDMRS effectively preserves important logical reasoning capabilities. Figure 6 shows consistent advantages are also observed in the tool-use domain. These results provide strong empirical validation for our theoretical analysis: the consistent advantage of EDMRS can be attributed to the issue of polysemantic experts. Experts selected under the AF or Full-parameter strategies are prone to cross-domain interference, which dilutes their effectiveness. In contrast, EDMRS explicitly mitigates this problem by imposing monosemanticity constraints, thereby promoting the selection of more specialized and reliable experts for the given task.

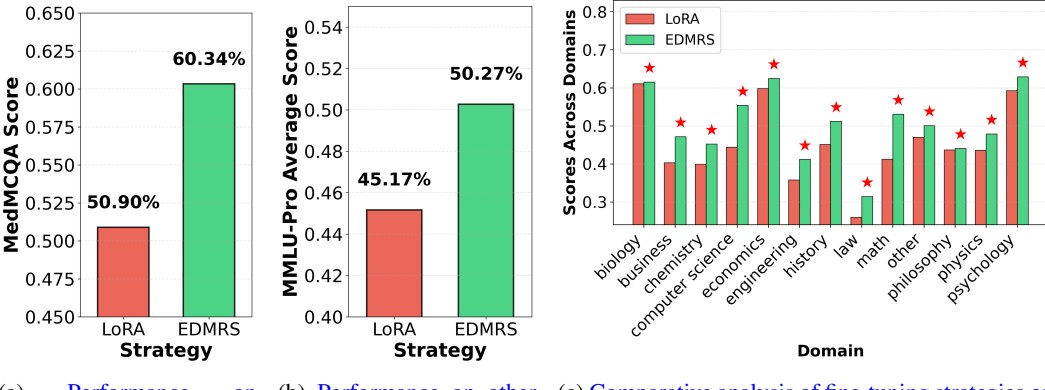

(a) Performance on Medicine Domain

(b) Performance on other domains

(c) Comparative analysis of fine-tuning strategies on out-of-domain tasks

Figure 7: Comparative analysis of EDMRS and LoRa across domains. Panel (a) evaluates medicine domain performance across strategies, Panel (b) measures cross-domain generalization on MMLU-Pro subsets, Panel (c) Comparative analysis of fine-tuning strategies on non-fine-tuned domains.

Our method is orthogonal to parameter-efficient fine-tuning approaches, such as applying LoRA's low-rank matrices directly to the experts we select. However, to compare their standalone capabilities, we evaluate against LoRA. As shown in Figure 7, EDMRS achieves significantly better performance on the target domain, while consistently outperforming LoRA across all non-target domains and on average, further demonstrating the advantages of EDMRS (Hu et al., 2022).

Taking into account both the comparable performance in the fine-tuned domain and the improvements on retention across other domains, the experimental results validate our core hypothesis: the EDMRS strategy consistently outperforms the activation frequency strategy and full-parameter fine-tuning. These findings demonstrate the substantial advantage of our monosemanticity-driven intra-layer expert selection approach.

## 6 CONCLUSION

This paper explores expert-level monosemanticity in MoE LLMs and proposes a fine-tuning strategy guided by monosemanticity analysis. We introduce the evaluation framework that extends SAE-based monosemanticity analysis to MoE architectures, highlighting the critical role of sparsity in promoting expert specialization across domains. To characterize expert behavior, we propose four novel metrics—EMS, LDS, EDMS, and EDMRS—that evaluate both monosemanticity and domain alignment. Building on these metrics, we develop a selective fine-tuning approach that leverages monosemantic scores to identify and update only the most relevant experts for the fine-tuned domain, while keeping unrelated or less critical experts frozen. Experiments show that our method achieves performance comparable to full-parameter fine-tuning within the target domain, while significantly enhancing performance retention and generalization across other domains. We believe that our framework provides a principled approach for understanding and controlling expert behavior in fine-tuning MoE LLMs, with potential applicability to a wide range of real-world scenarios.

ETHICS STATEMENT

This research strictly complies with the ICLR Code of Ethics (`https://iclr.cc/public/CodeOfEthics`). All experiments are conducted using publicly available datasets, including the MedMCQA dataset, the tool-use training dataset, the BFCL v3 tool-use test set, and the MMLU-Pro multi-domain test set. No personal data, sensitive information, or human subjects are involved at any stage of this work. The study does not contain any content that may raise ethical concerns, and there are no conflicts of interest, discrimination, or fairness issues. All authors have read and agreed to abide by the ICLR Code of Ethics. All data processing, model training, and evaluation procedures are performed in controlled environments to ensure compliance and transparency. The appendix provides detailed descriptions of data processing workflows, experimental settings, theoretical proofs, and prompt designs, further ensuring the integrity and traceability of this research. We welcome any questions or discussions regarding ethical considerations and will actively respond to concerns.

REPRODUCIBILITY STATEMENT

To ensure the reproducibility of our results, we provide comprehensive details of our experimental setup in Appendix A.6, including the base model (Qwen3-MoE-30B-A3B), the external LLM (GPT-4o), hardware environment (8 H20-141G GPUs), training hyperparameters (batch size, learning rate, warmup ratio, number of epochs, etc.), and the specific layers and number of experts selected for fine-tuning. All experiments are based on publicly available datasets, including the MedMCQA dataset, the tool-use training dataset, the BFCL v3 tool-use test set and the MMLU-Pro multi-domain test set. The definitions and calculation formulas for all metrics (EMS, EDMS, EDMRS, LDS) are provided in the main text and appendix, and the theoretical proof is detailed in Appendix A.4. The complete prompt designs used in the expert monosemanticity evaluation framework are presented in Appendix A.1, covering both explanation generation and evaluation stages. All experimental procedures and implementation details are thoroughly described in the main text and appendix, facilitating reproducibility and validation by the research community.

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

# A  APPENDIX

## A.1  DETAILS OF PROMPTS USED IN OUR MONOSEMANTICITY VALUATION FRAMEWORK

This section provides a detailed introduction to the prompt designs used in our expert monosemanticity valuation framework, covering both explanation generation and evaluation stages. The prompt contents are crucial for the accuracy of expert explanation generation and evaluation, and facilitate reproducibility and comparison.

### A.1.1  EXPLANATION GENERATION PROMPTS: $P_{mono}$ AND $P_{domain}^{(d)}$

During the explanation generation stage, we design two types of prompts: $P_{mono}$ and $P_{domain}^{(d)}$. Full details are as follows:

---

**Full Details of $P_{mono}$**

**System:**
We're studying experts in a large-scale Mixture of Experts (MoE) language model. Each expert specializes in processing certain types of content, knowledge domains, linguistic patterns, or reasoning tasks in documents. The content that activates each expert in documents is indicated with $<< ... >>$. We will give you a list of documents on which the expert is activated by the router, in order from most strongly activating to least strongly activating. Look at the parts of the document the expert activates for and summarize in a single sentence what type of content or domain this expert specializes in. Provide a reasonable explanation that captures the key distinguishing features. Note that some experts will specialize in very specific linguistic patterns or substrings, but others will activate on broader content types provided that the text contains particular knowledge domains or reasoning patterns. Your explanation should cover the general specialization pattern across most or all activating content (for example, don't give an explanation specific to a single domain if the expert activates across multiple related areas). Pay attention to things like the subject matter, linguistic style, technical complexity, or reasoning type of the activating content, if that seems relevant. Keep the explanation as short and simple as possible, limited to 20 words or fewer. Omit punctuation and formatting. You should avoid giving long lists of specific topics. Include specific details about the semantic patterns, syntactic structures, or functional roles that characterize this expert's specialization. Mention key distinguishing features such as specific word types, grammatical constructions, or conceptual themes.

Your response should be in the form "This expert specializes in...".

**User:**
Here is the explanation: this expert specializes in

Here are the examples:
{examples}

Here, {examples} refers to our sampled activation examples, with activated tokens marked using $<<>>$.

---

Full Details of $P_{domain}^{(d)}$

$P_{domain}^{(d)}$ includes all the content of $P_{mono}$. Additionally, to constrain $P_{domain}^{(d)}$ to generate explanations within a specific domain, taking the tool-use domain as an example, we add the following prompt on the basis of $P_{mono}$:

**System:**
For the medicine dataset:
This expert processes medical multiple-choice questions and clinical reasoning, so focus on medical terminology, diagnostic reasoning patterns, clinical decision-making, anatomical concepts, pharmacological knowledge, pathophysiology, treatment protocols, medical examination patterns, or healthcare domain knowledge.

IMPORTANT: This expert should ONLY specialize in medical and healthcare content. Do not generate explanations about non-medical domains.

For the tool-use dataset:
This expert processes tool usage and computational workflows. Focus on any specific aspect of tool interactions, including but not limited to: function invocations, parameter specifications, tool names, API endpoints, structured data formats, tool interaction patterns, command execution sequences, data transformation workflows, conditional logic in automation, error handling mechanisms, input/output processing, system integrations, or any other computational patterns within tool-calling contexts.

IMPORTANT: This expert can specialize in any granular aspect of tool usage, computational workflows, or function execution patterns. The specialization can be very specific (like particular parameter types, specific API patterns, or unique data structures) or broader (like general workflow patterns or interaction paradigms). Focus on the most distinctive characteristics that make this expert unique, even if they seem very specific or technical.

Here are diverse example patterns (but not limited to these): "This expert specializes in JSON parameter formatting in API calls", "This expert specializes in HTTP status code handling in tool responses", "This expert specializes in nested function call chains with dependency management", "This expert specializes in authentication token processing in API requests", "This expert specializes in error message parsing and retry logic", "This expert specializes in data type conversion between tool interfaces", "This expert specializes in asynchronous tool execution patterns", "This expert specializes in tool output validation and sanitization", "This expert specializes in command-line argument parsing and validation", "This expert specializes in database query construction in tool contexts", "This expert specializes in file path manipulation in system tools", "This expert specializes in regex pattern matching in data processing tools", "This expert specializes in timestamp and date formatting in tool outputs", "This expert specializes in memory management in computational workflows", "This expert specializes in parallel processing coordination in tool chains". Note that these are just examples - the expert may specialize in any specific, granular aspect of tool usage, function calls, or computational workflows.

Your response should be in the form "This expert specializes in...".

For the medicine dataset:
Some examples: "This expert specializes in differential diagnosis reasoning in multiple-choice contexts", "This expert specializes in pharmacological drug mechanism descriptions and interactions", "This expert specializes in anatomical structure identification and spatial relationships", "This expert specializes in clinical symptom pattern recognition and analysis", "This expert specializes in medical procedure descriptions and protocols", "This expert specializes in pathophysiology and disease mechanism explanations", "This expert specializes in medical terminology and nomenclature", "This expert specializes in treatment option

evaluation and selection".

For the tool-use dataset:
Some examples: "This expert specializes in processing rhetorical questions containing epistemic verbs", "This expert specializes in decision-making and preference-related reasoning", "This expert specializes in words with specific prefix patterns", "This expert specializes in government economic policy content".

## A.2 EXPLANATION EVALUATION PROMPTS

In the explanation evaluation stage, the full details of the designed prompts are as follows:

---

**Full Details of Explanation Evaluation Prompts**

**System:**
We're studying experts in a large-scale Mixture of Experts (MoE) language model. Each expert in the MoE layers specializes in processing certain types of content, knowledge domains, linguistic patterns, or reasoning tasks.

You will be given a short explanation of what this expert specializes in, and then be shown N example sequences in random order.

You will have to return a comma-separated list of the examples where you think this expert should be activated by the router, based on the content type, domain knowledge, or linguistic characteristics required.

For example, your response might look like "1, 3, 5".

Try not to be overly specific in your interpretation of the explanation.

If you think there are no examples where this expert will be significantly activated, you should just respond with "None".

You should include nothing else in your response other than comma-separated numbers or the word "None" - this is important.
**User:**

The documents which this expert is activated are given below:
{examples}
Here, {examples} is a set of numbered samples, where each sample is shown in its original form without <<>> indicating activated tokens.

---

## A.3 EXPERT EXPLANATIONS FROM THE MONOSEMANTICITY VALUATION FRAMEWORK

To enhance the understanding of expert specialization within MoE, we present the natural language explanations generated by our monosemanticity valuation framework below.

> **General Expert Explanations by $\mathrm{P}_{mono}$**
>
> - This expert specializes in processing geographical coordinates and East Asian country names and language codes.
> - This expert specializes in processing scientific and mathematical content such as synaptic input, quantum physics, and neuronal activity rate.
> - This expert specializes in content related to gas and electricity prices and environmental impact on air quality.
> - This expert specializes in documentation or metadata with type declarations and default value specifications.
> - This expert specializes in processing scientific measurements and data conversion parameters in technical domains.

> **Tool-Use Domain Expert Explanations by $\mathrm{P}_{domain}^{(d)}$**
>
> - This expert specializes in tool usage and API function calls with parameter specifications and structured interaction patterns.
> - This expert specializes in binary calculation tool interactions including parameters and function calls for binary operations.
> - This expert specializes in tool usage patterns including API function invocations, parameter specifications, and structured data formats.
> - This expert specializes in tool invocation patterns and response handling with structured data formats.
> - This expert specializes in the structuring and handling of geographic coordinates and phone numbers within data formats.

### A.4 DETAILED PROOF OF THEOREM 1

The detailed proof of Theorem 1 is provided below.

*Proof.* By the law of total expectation, the expected total performance change for expert $E_i$ across all domains is given by:

$$\mathbb{E}[C_i] = \mathbb{E}[C_i(t)]$$
$$= \mathbb{E}[\Delta_{\mathcal{P}}^{(d)}(E_i, t)] - \mathbb{E}[\Delta_{\mathcal{N}}^{(\text{other})}(E_i, t)].$$

Within the sparse activation mechanism of the MoE architecture, an expert only contributes to the model output when it is activated. If the expert is not activated, it does not induce any performance change, i.e.,

$$\mathbb{E}[\Delta_{\mathcal{P}}^{(d)}(E_i, t) \mid E_i''] = 0 \tag{11}$$
$$\mathbb{E}[\Delta_{\mathcal{N}}^{(\text{other})}(E_i, t) \mid E_i''] = 0 \tag{12}$$

where $E_i''$ indicates that expert $E_i$ is not activated during the output of token $t$.

Therefore, by the law of total expectation, the expectation can be decomposed into the cases of activation and non-activation:

$$\mathbb{E}[\Delta_{\mathcal{P}}^{(d)}(E_i, t)] = P(E_i') \cdot \mathbb{E}[\Delta_{\mathcal{P}}^{(d)}(E_i, t) \mid E_i']$$
$$+ P(E_i'') \cdot \mathbb{E}[\Delta_{\mathcal{P}}^{(d)}(E_i, t) \mid E_i'']$$
$$= \mathrm{AF}_i^{(d)} \cdot \mathbb{E}[\Delta_{\mathcal{P}}^{(d)}(E_i, t) \mid E_i'] + (1 - \mathrm{AF}_i^{(d)}) \cdot 0 \tag{13}$$
$$= \mathrm{AF}_i^{(d)} \cdot \mathbb{E}[\Delta_{\mathcal{P}}^{(d)}(E_i, t) \mid E_i']$$

Similarly,

$$\mathbb{E}[\Delta_{\mathcal{N}}^{(\text{other})}(E_i, t)] = \mathrm{AF}_i^{(d)} \cdot \mathbb{E}[\Delta_{\mathcal{N}}^{(\text{other})}(E_i, t) \mid E_i'] \tag{14}$$

Thus, the expected total performance change is given by:

$$\mathbb{E}[C_i] = \text{AF}_i^{(d)} \cdot \left( \mathbb{E}[\Delta_{\mathcal{P}}^{(d)}(E_i, t) \mid E_i'] - \mathbb{E}[\Delta_{\mathcal{N}}^{(\text{other})}(E_i, t) \mid E_i'] \right) \tag{15}$$

Therefore, the expected total performance change can be approximated as:

$$\mathbb{E}[C_i] = \text{AF}_i^{(d)} \left( k_1 \, \text{EDMS}_i^{(d)} - k_2 \, (1 - \text{EDMS}_i^{(d)}) \right)$$

$$= \text{AF}_i^{(d)} \left[ (k_1 + k_2) \, \text{EDMS}_i^{(d)} - k_2 \right]$$

Our objective is to select an expert set S that maximizes the expected total performance change after fine-tuning in the fine-tuned domain $d$, i.e.,

$$\text{S} = \arg \max_{|\text{S}|=m} \sum_{i \in \text{S}} \mathbb{E}[C_i]$$

Since $k_1$ and $k_2$ are constants, they do not affect the ranking of expert selection. Thus, the optimization objective is equivalent to:

$$\text{S} = \arg \max_{|\text{S}|=m} \sum_{i \in \text{S}} \mathbb{E}[C_i]$$

$$= \arg \max_{|\text{S}|=m} \sum_{i \in \text{S}} \text{EDMRS}_i^{(d)}$$

In other words, to maximize the total EDMRS score, the $m$ experts with the highest EDMRS scores should be selected, which leads to our EDMRS strategy and demonstrates its optimality.

$\square$

## A.5 STABILITY OF METRICS WITH AFFORDABLE EXTERNAL LLMS

To assess the stability of monosemanticity metrics when using more cost-effective external LLMs, we conducted experiments on Qwen3-235B-A22B using both GPT-4o and the smaller, less expensive GPT-4o-mini model to evaluate all layers under $\text{P}_{mono}$ and $\text{P}_{domain}^{(d)}$ prompts. The comparative results are shown in Figure 8, where LDDS represents the average of EDMS scores, reflecting the stability of EDMS—analogous to how LDS reflects the stability of EMS. Notably, the relative variation trends are highly consistent across different external LLMs, and the results also clearly show the phenomenon of EDMS being smaller than EMS, which again indicates that expert features are superimposed. Therefore, even when switching to a more affordable external LLM, our scoring remains highly reliable.

## A.6 EXPERIMENTS SETTINGS

In our experiments, GPT-4o is adopted as the external LLM within the evaluation framework, while Qwen3-MoE-30B-A3B serves as the base model. All training and validation procedures are performed on a server equipped with 8 H20-141G GPUs. For the comparative experiments on the MedMCQA dataset with three strategies, the hyperparameter configurations are as follows: we use a batch size of 32 and a learning rate of 5e-5. For the tool-use dataset, all methods utilize a batch size of 128, cosine annealing learning rate scheduling, an initial learning rate of 1e-5, a warmup ratio of 0.05, and a minimum learning rate of 1e-6. The loss weight for load balancing is set to 5e-2. Due to the difference in the number of parameters updated at each training step, the EDMRS and AF models are trained for 3 epochs, while the Full-parameter model is trained for 1 epoch. When applying the expert selection fine-tuning strategy, we select all layers of the model and only fine-tune 16 experts within each layer.

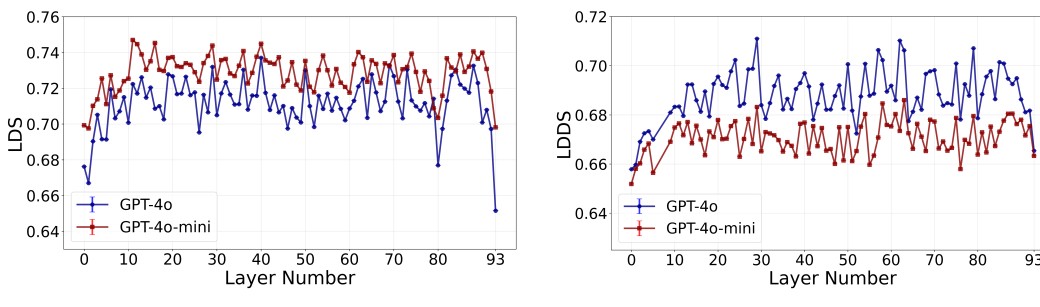

(a) Comparison of LDS Values across Different Layers

(b) Comparison of LDDS Values across Different Layers

Figure 8: Analysis of metric stability under affordable external LLMs. Panel (a) shows the trend of LDS scores across different layers for various external LLMs. Panel (b) shows the trend of LDDS scores across different layers for various external LLMs.

## A.7 USE OF LARGE LANGUAGE MODELS

The present manuscript's Introduction and Conclusion underwent limited language refinement through a general-purpose large language model, strictly for improving grammar, wording, and readability. The model played no role in conceptualization, data or code generation, image production, mathematical derivation, data analysis, literature search, or citation management. Its use was confined to light textual polishing of pre-written content, with no confidential or external data disclosed. All AI-proposed changes were carefully examined and manually approved by the authors, who retain sole accountability for the scholarly content of the work.

