# OpenReview forum: "Understanding and Leveraging Expert-level Monosemanticity in Finetuning MoE LLMs"
_ICLR.cc/2026/Conference — ICLR 2026 Conference Withdrawn Submission_

### Official Review · Reviewer_5HY4 · 2025-10-25

**Soundness:** 1
**Presentation:** 2
**Contribution:** 2
**Rating:** 2
**Confidence:** 3

**Summary:**

The paper proposes a framework to evaluate expert-level monosemanticity in Mixture-of-Experts (MoE) language models, inspired by SAE. It defines new metrics (EMS, EDMS, LDS, EDMRS) to quantify monosemanticity and uses them to guide selective fine-tuning of domain-relevant experts, aiming to preserve cross-domain generalization. Experiments on tool-use data suggest that monosemanticity-guided tuning achieves similar in-domain performance as full fine-tuning while reducing degradation on other domains.

**Strengths:**

1. The paper tackles an important and timely topic, interpretability and efficient adaptation of MoE LLMs.

2. The focus on explainability and expert-level interpretability is valuable and well-motivated, as it attempts to bridge the gap between activation analysis and human-understandable reasoning about experts.

**Weaknesses:**

1. The proposed metrics (EMS and EDMS) do not convincingly measure monosemanticity. They evaluate how accurately a separate LLM can predict expert activations from textual explanations, which captures predictability or consistency rather than semantic purity. Therefore, these scores cannot directly indicate whether an expert encodes a single coherent concept. All subsequent analyses and the EDMRS-based fine-tuning strategy are built upon this assumption, raising concerns about theoretical soundness.

2. The paper conflates semantic monosemanticity with functional capability. The concept of monosemanticity concerns the disentanglement of semantic features or conceptual meanings, but the chosen evaluation domain—tool use—reflects a procedural ability, not a semantic category. As a result, the study does not genuinely assess semantic interpretability, and further experiments across semantic or linguistic dimensions are necessary to substantiate the claim.

3. The reasoning in Insight 1 about sparsity and monosemanticity is weak. The authors claim that MoE’s sparse routing inherently increases monosemanticity compared to dense models, but they provide neither a controlled comparison with dense LLMs nor evidence of causality. The inference that “higher sparsity leads to stronger monosemanticity” is speculative and lacks rigorous support.

4. The methodological contribution is incremental. The framework remains fundamentally activation-based; it interprets standard activation statistics with natural-language summaries from an external LLM. While this adds a layer of explainability, it does not introduce new interpretability mechanisms or learning principles. The novelty lies primarily in presentation rather than substance.

5. The experiments are limited in scope and impact. The reported improvement (0.8%) on out-of-domain benchmarks is marginal, and the experiments cover only one task domain without ablation studies or statistical variance analysis. The results, while suggestive, are insufficient to establish the method’s effectiveness or general applicability.

**Questions:**

See weaknesses.

---

> ### Author Response · Authors · 2025-11-28
> **Response to Reviewer 5HY4 (part1)**
>
> We sincerely thank you for your constructive feedback and deep insights. We are greatly encouraged by your assessment that our topic is "important, timely, valuable, and well-motivated." Our research contributes not only by proposing a fine-tuning strategy to mitigate catastrophic forgetting but also by advancing the understanding of MoE interpretability, which remains underexplored despite widespread use. You raised concerns that our metrics may not convincingly measure semantic monosemanticity and that semantic monosemanticity might be conflated with functional capability. We address each of your points below and hope to fully resolve your concerns.
>
> ### W1: The proposed metrics (EMS and EDMS) do not convincingly measure monosemanticity
> Our metrics are inspired by existing research in the field of Sparse Autoencoder (SAE) monosemanticity. Substantial research has demonstrated that this type of automated interpretability analysis effectively measures monosemanticity, rather than merely reflecting predictability or consistency [1].
>
> ### W2: Tool use reflects a procedural ability, not a semantic category
> Regarding your point that tool use reflects procedural ability rather than a semantic category, we acknowledge that while the tool-use domain possesses some semantic aspects, it is not purely semantic, which may have contributed to the modest experimental gains in our initial results [2]. To address this, we introduced the new **MedMCQA** dataset to execute our full pipeline—from MoE interpretability analysis to guided fine-tuning—applying our strategy to all layers. MedMCQA is a large-scale medical dataset with 183k training samples, evaluated on its official validation set [3]. For cross-domain testing, we used **13 distinct non-health subsets of MMLU-Pro: biology, business, chemistry, computer science, economics, engineering, history, law, math, other, philosophy, physics, and psychology**, ensuring comprehensive coverage.
>
> The table below shows the accuracy of the three methods (Full, AF, EDMRS) across these 13 domain test sets:
> | Method | biology | business | chemistry | computer science | economics | engineering | history | law | math | other | philosophy | physics | psychology | MMLU-Pro Avg |
> | --- | --- | --- | --- | --- | --- | --- | --- | --- | --- | --- | --- | --- | --- | --- |
> | Full | 0.4686 | 0.3853 | 0.4364 | 0.2683 | 0.5284 | 0.3664 | 0.3911 | 0.2997 | 0.3494 | 0.3874 | 0.4148 | 0.4442 | 0.6529 | 0.4148 |
> | AF | 0.6039 | 0.4284 | 0.4408 | 0.4951 | 0.5711 | 0.3736 | 0.4094 | 0.2852 | 0.4796 | 0.4773 | 0.3968 | 0.4742 | 0.5050 | 0.4570 |
> | EDMRS | **0.6151** | **0.4715** | **0.4523** | **0.5537** | **0.6244** | **0.4118** | **0.5118** | **0.3143** | **0.5307** | **0.5011** | **0.4409** | **0.4788** | 0.6291 | **0.5027** |
> | EDMRS vs Full | **+31.26%** | **+22.37%** | **+3.64%** | **+106.37%** | **+18.17%** | **+12.39%** | **+30.86%** | **+4.87%** | **+51.89%** | **+29.35%** | **+6.29%** | **+7.79%** | -3.65% | **+21.19%** |
> | EDMRS vs AF | **+1.85%** | **+10.06%** | **+2.61%** | **+11.84%** | **+9.33%** | **+10.22%** | **+25.01%** | **+10.20%** | **+10.65%** | **+4.99%** | **+11.11%** | **+0.97%** | **+24.57%** | **+10.02%** |
>
> Meanwhile, our strategy achieves performance on the target domain extremely close to the Full strategy (0.6034 vs. 0.6070, a difference of only 0.6%) and significantly outperforms the AF strategy (0.6034 vs. 0.4848, exceeding it by 24.46%). Thus, the results fully and significantly demonstrate the effectiveness of our method: while matching or surpassing the baselines in the target domain, our method achieves the best results in 12 out of 13 other domain test sets. On average, **our method outperforms AF by 10.02% and Full by 21.19%**. Encouragingly, **EDMRS outperforms Full by 106.37% in computer science and 51.89% in math**, reflecting that our method fully preserves important logical reasoning capabilities. The reason for AF's poor performance across both target and other domains is exactly as described in our paper: it selects "generalist" experts responsible for other domains rather than true "specialists" for the current domain, leading to severe catastrophic forgetting during training. We hope this addresses your concern.

---

> ### Author Response · Authors · 2025-11-28
> **Response to Reviewer 5HY4 (part2)**
>
> ### W3: The reasoning in Insight 1 about sparsity and monosemanticity.
> Our proposed Insight 1 is primarily based on the substantial existing research in the field of Sparse Autoencoders (SAE), where the core finding is that sparsity enhances monosemanticity, and our LDS metric reflects the strength of this monosemanticity [1][4][5]. For Sparse MoE, its sparse activation characteristic means that non-activated parts are effectively zero, resulting in a highly sparse overall augmented output vector; a smaller Top-k implies greater sparsity. In Figure 2a, we observe that as sparsity increases, the LDS score increases—indicating enhanced monosemanticity—which serves as the experimental basis for Insight 1. Our analysis framework is built on acquiring sparse features via Top-k activation to leverage the extensive research in the SAE field. As $k$ increases, the MoE model increasingly approximates a dense model, and monosemanticity deteriorates; this reflects the monosemanticity condition of dense models, thereby allowing us to deduce the correlation between sparsity and monosemanticity. We hope this explanation clearly elucidates the deductive logic of Insight 1 and addresses your query.
>
> ### W4: The framework remains fundamentally activation-based
>
> To address the distinction between monosemanticity and activation frequency, we first provide a motivational explanation and then demonstrate the validity of our explanation through the significant experimental performance difference between EDMRS and AF in our new experiments.
>
> **Motivation:** The activation frequency strategy selects experts with high activation rates. However, these experts are often "generalists" responsible for broad capabilities, causing them to activate across multiple vertical domains. Fine-tuning these experts damages their original general capabilities. In contrast, EDMRS reduces the selection weight of these generalists, ensuring that the experts we select are genuinely aligned with the semantics of the current domain and play a critical role. Therefore, compared to the AF strategy, our method causes less damage to experts required by other domains and effectively fine-tunes experts that truly correspond to the current domain's semantics.
>
> **Experimental Verification:** As mentioned above, the new experimental results significantly demonstrate the effectiveness of our method. While matching or surpassing Full and AF in the target domain, our method achieves the best results in 12 of the 13 other domain test sets. On average, **our method outperforms AF by 10.02% and Full by 21.19%**. Encouragingly, **EDMRS surpasses Full by 106.37% in computer science and 51.89% in math**, reflecting that our method fully preserves important logical reasoning capabilities. The reason for the AF strategy's poor performance across both target and other domains is exactly as described in our paper: it selects "generalist" experts responsible for other domains rather than true "specialists" for the current domain, leading to severe catastrophic forgetting during training.
>
> We hope this resolves your doubts regarding the distinction between monosemanticity and activation frequency.
>
> ### W5: The reported improvement (0.8%) on out-of-domain benchmarks is marginal
> As noted above, we addressed this by introducing the new dataset and extending our strategy to all 48 layers. Our strategy achieves target domain performance comparable to Full (0.6034 vs. 0.6070, a 0.6% gap) and significantly outperforms AF (0.6034 vs. 0.4848, +24.46%), while **outperforming AF by 10.02% and Full by 21.19%** on MMLU-Pro. Encouragingly, **EDMRS surpasses Full by 106.37% in computer science and 51.89% in math**, demonstrating substantial preservation of critical logical reasoning capabilities.
>
> ---
>
> We hope these responses fully address your concerns. We sincerely thank you again for your valuable feedback.
>
> [1] Gonc¸alo Paulo, Alex Mallen, Caden Juang, and Nora Belrose. Automatically interpreting millions of features in large language models, 2024. URL [https://arxiv.org/abs/2410.13928.](https://arxiv.org/abs/2410.13928.)
>
> [2] Galichin A, Dontsov A, Druzhinina P, et al. I Have Covered All the Bases Here: Interpreting Reasoning Features in Large Language Models via Sparse Autoencoders[J]. arXiv preprint arXiv:2503.18878, 2025.
>
> [3] Pal A, Umapathi L K, Sankarasubbu M. MedMCQA: A large-scale multi-subject multi-choice dataset for medical domain question answering[C]//Conference on health, inference, and learning. PMLR, 2022: 248-260.
>
> [4] Hoagy Cunningham, Aidan Ewart, Logan Riggs, Robert Huben, and Lee Sharkey. Sparse autoencoders find highly interpretable features in language models. arXiv preprint arXiv:2309.08600, 2023[.](https://arxiv.org/abs/2410.13928.)
>
> [5] Leo Gao, Tom Dupre la Tour, Henk Tillman, Gabriel Goh, Rajan Troll, Alec Radford, Ilya ´ Sutskever, Jan Leike, and Jeffrey Wu. Scaling and evaluating sparse autoencoders. arXiv preprint arXiv:2406.04093, 2024.

---

### Official Review · Reviewer_RmuA · 2025-10-29

**Soundness:** 3
**Presentation:** 2
**Contribution:** 2
**Rating:** 6
**Confidence:** 3

**Summary:**

This paper studies interpretability and fine-tuning in Mixture-of-Experts (MoE) large language models through expert-level monosemanticity, where each expert represents a distinct concept. Inspired by sparse autoencoders, the authors propose quantitative metrics (EMS, EDMS) to measure monosemanticity and introduce a selective fine-tuning strategy based on the Expert Domain Monosemantic Responsibility Score (EDMRS). The method fine-tunes only domain-relevant experts, reducing interference across domains. Experiments on tool-use tasks show comparable in-domain results to full tuning while improving cross-domain generalization, offering an interpretable and efficient framework for MoE adaptation.

**Strengths:**

1. This paper presents a clear theoretical framework with formal derivations.

2. This paper demonstrates practical improvements in maintaining task performance and enhancing cross-domain generalization.

**Weaknesses:**

1. Limited experimental scope: Evaluation is restricted to a single tool-use task, lacking validation on more challenging language understanding or reasoning tasks.

2. Dependence on external LLM (GPT-4o) for explanation generation introduces potential evaluation bias and high computational cost.

3. Strong theoretical assumptions: The hypothesized linear relationship between EDMS and performance improvement is not empirically substantiated.

4. Missing comparisons with mainstream fine-tuning methods such as LoRA, Adapter, or Router-tuning.

5. Small performance gain (≈0.8%), which may not convincingly demonstrate general superiority.

**Questions:**

1. How stable are EMS and EDMS results when using different external LLMs?

2. Have the authors considered validating EDMRS effectiveness in multi-task or multi-domain fine-tuning scenarios?

3. Could a lighter-weight interpretability proxy model replace GPT-4o for explanation generation?

4. Could EDMRS be combined with parameter-efficient fine-tuning methods (e.g., LoRA or Adapter) to enhance generality?

---

> ### Author Response · Authors · 2025-11-28
> **Response to Reviewer RmuA (part1)**
>
> We sincerely thank you for your constructive feedback and insightful comments. We are greatly encouraged by your recognition of our clear theoretical framework, formal derivations, and the practical improvements demonstrated by our method. Our study contributes not only by proposing a fine-tuning strategy to mitigate catastrophic forgetting but also by aiding the understanding of MoE interpretability, an area widely applied yet underexplored. You noted concerns regarding limited scope and the evaluation bias/cost of relying on external LLMs. We address each of your points below and hope to fully resolve your concerns.
>
> ### W1: Limited experimental scope
> We expanded our scope by adding the **MedMCQA** dataset to execute our full pipeline—from interpretability analysis to guided fine-tuning—applying our strategy to all layers. MedMCQA is a large-scale medical dataset with 183k training samples, evaluated on its official validation set [1]. For cross-domain testing, we used **13 distinct non-health subsets of MMLU-Pro: biology, business, chemistry, computer science, economics, engineering, history, law, math, other, philosophy, physics, and psychology**, ensuring comprehensive coverage.The table below shows the accuracy of the three methods (Full, AF, EDMRS) across these 13 domain test sets:
> | Method | biology | business | chemistry | computer science | economics | engineering | history | law | math | other | philosophy | physics | psychology | MMLU-Pro Avg |
> | --- | --- | --- | --- | --- | --- | --- | --- | --- | --- | --- | --- | --- | --- | --- |
> | Full | 0.4686 | 0.3853 | 0.4364 | 0.2683 | 0.5284 | 0.3664 | 0.3911 | 0.2997 | 0.3494 | 0.3874 | 0.4148 | 0.4442 | 0.6529 | 0.4148 |
> | AF | 0.6039 | 0.4284 | 0.4408 | 0.4951 | 0.5711 | 0.3736 | 0.4094 | 0.2852 | 0.4796 | 0.4773 | 0.3968 | 0.4742 | 0.5050 | 0.4570 |
> | EDMRS | **0.6151** | **0.4715** | **0.4523** | **0.5537** | **0.6244** | **0.4118** | **0.5118** | **0.3143** | **0.5307** | **0.5011** | **0.4409** | **0.4788** | 0.6291 | **0.5027** |
> | EDMRS vs Full | **+31.26%** | **+22.37%** | **+3.64%** | **+106.37%** | **+18.17%** | **+12.39%** | **+30.86%** | **+4.87%** | **+51.89%** | **+29.35%** | **+6.29%** | **+7.79%** | -3.65% | **+21.19%** |
> | EDMRS vs AF | **+1.85%** | **+10.06%** | **+2.61%** | **+11.84%** | **+9.33%** | **+10.22%** | **+25.01%** | **+10.20%** | **+10.65%** | **+4.99%** | **+11.11%** | **+0.97%** | **+24.57%** | **+10.02%** |
>
> Meanwhile, our strategy achieves performance on the target domain extremely close to the Full strategy (0.6034 vs. 0.6070, a difference of only 0.6%) and significantly outperforms the AF strategy (0.6034 vs. 0.4848, exceeding it by 24.46%).Thus, the results significantly demonstrate the effectiveness of our method. While matching or surpassing Full and AF in the target domain, our method achieves the best results in 12 of the 13 other domains. On average, **it outperforms AF by 10.02% and Full by 21.19%**. Encouragingly, **EDMRS surpasses Full by 106.37% in computer science and 51.89% in math**, reflecting strong preservation of logical reasoning capabilities. The reason for AF's poor overall performance is, as discussed in our paper, that it selects "generalist" experts responsible for other domains rather than true "specialists," causing severe catastrophic forgetting. We hope this addresses your query.
>
> ### W2: Dependence on external LLM (GPT-4o), introduces potential evaluation bias and high computational cost
> We evaluated using the smaller, more cost-effective gpt-4o-mini. The results show highly consistent relative trends in scores between the two models despite the significant price difference. This demonstrates the stability of our method and the feasibility of using cheaper models to reduce costs. Detailed results are in Figure 8 (page 18) of the revised paper.
>
> ### W3: Strong theoretical assumptions
> As you noted, clear formal derivations require assumptions. Our logic posits that an expert with a high EDMS has activation patterns highly consistent with the domain explanation, implying alignment with target domain features. Thus, we expect it to contribute more to domain performance while minimizing interference with other domains due to functional decoupling. We validate this assumption by evaluating the EDMRS method built upon it. We believe our significant empirical results provide reverse validation for our theoretical assumption's rationality: **our strategy matches Full performance in the target domain (0.6034 vs. 0.6070, -0.6%) and beats AF (+24.46%), while outperforming AF by 10.02% and Full by 21.19% on MMLU-Pro. Notably, EDMRS surpasses Full by 106.37% in computer science and 51.89% in math**, confirming substantial preservation of critical logical reasoning capabilities.

---

> ### Author Response · Authors · 2025-11-28
> **Response to Reviewer RmuA (part2)**
>
> ### W4: Missing comparisons with mainstream fine-tuning methods
> First, due to the structural properties of MoE expert networks and our design, our method is orthogonal to, rather than mutually exclusive with, these fine-tuning methods. Our approach essentially selects a subset of parameters that are relevant to the target domain and decoupled from others; existing methods can then treat these selected parameters as a new neural network for direct application (e.g., applying LoRA's low-rank matrices). However, to address your query, we conducted the following experiment comparing EDMRS with LoRA:
> | Method | MedMCQA(target domain) | biology | business | chemistry | computer science | economics | engineering | history | law | math | other | philosophy | physics | psychology | MMLU-Pro Avg |
> | --- | --- | --- | --- | --- | --- | --- | --- | --- | --- | --- | --- | --- | --- | --- | --- |
> | LoRA | 0.5090 | 0.6109 | 0.4030 | 0.3993 | 0.4439 | 0.5983 | 0.3581 | 0.4514 | 0.2598 | 0.4123 | 0.4697 | 0.4369 | 0.4357 | 0.5927 | 0.4517 |
> | EDMRS | **0.6034** | **0.6151** | **0.4715** | **0.4523** | **0.5537** | **0.6244** | **0.4118** | **0.5118** | **0.3143** | **0.5307** | **0.5011** | **0.4409** | **0.4788** | **0.6291** | **0.5027** |
> | EDMRS vs LoRA | **+18.55%** | **+0.69%** | **+17.00%** | **+13.27%** | **+24.74%** | **+4.36%** | **+15.00%** | **+13.38%** | **+20.98%** | **+28.12%** | **+6.69%** | **+0.92%** | **+9.89%** | **+6.14%** | **+11.30%** |
>
> The results show that **EDMRS significantly outperforms the classic LoRA algorithm, achieving superior performance in the target domain (+18.55%) while maintaining a clear lead in other domains (+11.3% on average)**. This demonstrates the superiority of our method. We hope this answers your question.
>
> ### W5: Small performance gain (≈0.8%)
> As noted above, we addressed this by introducing the new dataset and extending our strategy to all 48 layers. Our strategy achieves target domain performance comparable to Full (0.6034 vs. 0.6070, a 0.6% gap) and significantly outperforms AF (0.6034 vs. 0.4848, +24.46%), while **outperforming AF by 10.02% and Full by 21.19%** on MMLU-Pro. Encouragingly, **EDMRS surpasses Full by 106.37% in computer science and 51.89% in math**, demonstrating substantial preservation of critical logical reasoning capabilities.
> ### Q1: How stable are EMS and EDMS results when using different external LLMs
> We evaluated using the smaller, more cost-effective gpt-4o-mini. Results show highly consistent relative trends in scores between the two models despite the price difference, demonstrating the stability of our method and the feasibility of using cheaper models to reduce costs. See Figure 8 (page 18) in the paper for details.
> ### Q2: Validating EDMRS effectiveness in multi-task or multi-domain
> As mentioned above, we addressed this by adding the new dataset and obtaining significant experimental results.
>
> ### Q3: Could a lighter-weight interpretability proxy model replace GPT-4o?
> We evaluated using the smaller, more cost-effective **gpt-4o-mini**. Results show highly consistent relative trends in scores between the two models despite the price difference, demonstrating the stability of our method and the feasibility of using cheaper models to reduce costs. See Figure 8 (page 18) in the paper for details.
>
> ### Q4: Could EDMRS be combined with parameter-efficient fine-tuning methods
> Thanks to the structural properties of MoE expert networks and our design, our method is orthogonal to—rather than mutually exclusive with—these fine-tuning methods. Our approach essentially identifies a subset of parameters that are relevant to the target domain and decoupled from others; PEFT methods can then be directly applied to these selected parameters (e.g., applying LoRA's low-rank matrices to the selected experts).
>
> ---
>
> We hope these responses fully address your concerns. We sincerely thank you again for your valuable suggestions.
>
> [1] Pal A, Umapathi L K, Sankarasubbu M. MedMCQA: A large-scale multi-subject multi-choice dataset for medical domain question answering[C]//Conference on health, inference, and learning. PMLR, 2022: 248-260.

---

### Official Review · Reviewer_4Z84 · 2025-10-31

**Soundness:** 2
**Presentation:** 2
**Contribution:** 2
**Rating:** 2
**Confidence:** 4

**Summary:**

This paper explores the interpretability of experts (such as specialization for different domains) in the Mixture-of-Expert (MoE) architecture. To this end, authors analyse monosemantic representations in MoE experts. They find that increased sparsity in MoEs improves expert-level monosemantic representations, hence the specialization across downstream domains. Based on their analysis, the paper proposes a specialized supervised finetuning method for MoEs where "selected" experts are updated during finetuning. To select these experts for a domain, they propose using monosemanticity scores combined with activation frequency. Their results on a tool-use dataset show that while this method achieves comparable performance with full model finetuning, it maintains the performance in the other domains.

**Strengths:**

1. Interpretability analysis for the MoE expert is a highly important topic. Although recent LLMs are adapting the MoE architecture, expert specialization and exploration on this is commonly underexplored.

2. The paper proposes an evaluation framework for MoE models to examine monosemanticity. They use the output of the MoE gate, similar to the hidden representation in SAE.

**Weaknesses:**

1. Gaps in monosemanticity analysis:

a. It is not very clear how the monosemanticity evaluation framework is fundamentally different than analysing expert activation frequency on sampled datasets.

b. The MoE models that were experimented with have been trained with a certain number of activated experts during their pretraining. Without having control over this, varying the activated expert during the test time may lead incomplete conclusion. For this, LLMs with different pretraining sparsity needs to be explored.

2. Experiments include only one dataset (BFCL v3, tool-use) as the ownstream domain. From this experiment, it is not possible say that the findings are generalizable across different domains or an inherent property of this particular domain.

2. Results compared to the baselines are within 1%. Although this does not necessarily invalidate the results, statistical analysis and multiple runs with different seeds are required to assess the conclusion.

**Questions:**

1. Why do the finetuning experiments only include a single layer?

2. The selected model is trained with 8 expert activated per token; any fine-tuning changing this strategy may give suboptimal results. Why were 16 experts selected to update?

---

> ### Author Response · Authors · 2025-11-28
> **Response to Reviewer 4Z84 (part1)**
>
> We sincerely thank you for your constructive feedback and for recognizing our work as a highly important topic. Beyond addressing catastrophic forgetting, our study advances the understanding of MoE interpretability. Below, we address your specific concerns regarding the distinction from Activation Frequency (AF), test-time expert activation, and dataset diversity, aiming to fully resolve them.
>
> ### W1: Gaps in monosemanticity analysis
> ### w1.a how the monosemanticity evaluation framework is fundamentally different than analysing expert activation frequency on sampled datasets
> To address the distinction between monosemanticity and activation frequency, we first explain our motivation and then validate it through the significant performance difference between EDMRS and AF in our new experiments.
>
> **Motivation:** The AF strategy selects experts based on high activation rates. However, these experts are often "generalists" with broad capabilities, activating across multiple domains. Fine-tuning them for a specific task causes severe damage to their general abilities. In contrast, EDMRS reduces the selection weight of these generalists, ensuring we select experts that are genuinely semantically aligned with the target domain. Consequently, compared to AF, our method minimizes harm to experts needed for other domains while focusing training on experts truly relevant to the current domain's semantics.
>
> **New Experimental Evidence:** The significant results from our new experiments strongly validate our explanation. We introduced the **MedMCQA** dataset to execute our full pipeline—from MoE interpretability analysis to guided fine-tuning—applying our strategy to **all layers**. MedMCQA is a large-scale medical dataset with 183k training samples, and we evaluated medical capability on its official validation set [1]. For cross-domain evaluation, we tested on **13 distinct non-health subsets of MMLU-Pro: biology, business, chemistry, computer science, economics, engineering, history, law, math, other, philosophy, physics, and psychology**, ensuring a comprehensive assessment.
>
> The table below shows the accuracy of three methods (Full, AF, EDMRS) across these 13 domains:
>
> | Method | biology | business | chemistry | computer science | economics | engineering | history | law | math | other | philosophy | physics | psychology | MMLU-Pro Avg |
> | :--- | :--- | :--- | :--- | :--- | :--- | :--- | :--- | :--- | :--- | :--- | :--- | :--- | :--- | :--- |
> | Full | 0.4686 | 0.3853 | 0.4364 | 0.2683 | 0.5284 | 0.3664 | 0.3911 | 0.2997 | 0.3494 | 0.3874 | 0.4148 | 0.4442 | 0.6529 | 0.4148 |
> | AF | 0.6039 | 0.4284 | 0.4408 | 0.4951 | 0.5711 | 0.3736 | 0.4094 | 0.2852 | 0.4796 | 0.4773 | 0.3968 | 0.4742 | 0.5050 | 0.4570 |
> | EDMRS | **0.6151** | **0.4715** | **0.4523** | **0.5537** | **0.6244** | **0.4118** | **0.5118** | **0.3143** | **0.5307** | **0.5011** | **0.4409** | **0.4788** | 0.6291 | **0.5027** |
> | EDMRS vs Full | **+31.26%** | **+22.37%** | **+3.64%** | **+106.37%** | **+18.17%** | **+12.39%** | **+30.86%** | **+4.87%** | **+51.89%** | **+29.35%** | **+6.29%** | **+7.79%** | -3.65% | **+21.19%** |
> | EDMRS vs AF | **+1.85%** | **+10.06%** | **+2.61%** | **+11.84%** | **+9.33%** | **+10.22%** | **+25.01%** | **+10.20%** | **+10.65%** | **+4.99%** | **+11.11%** | **+0.97%** | **+24.57%** | **+10.02%** |
>
> Meanwhile, our strategy achieves performance on the target domain extremely close to the Full strategy (0.6034 vs. 0.6070, a difference of only 0.6%) and significantly outperforms the AF strategy (0.6034 vs. 0.4848, exceeding it by 24.46%). Thus, the results fully and significantly demonstrate the effectiveness of our method: while matching or surpassing the baselines in the target domain, our method achieves the best results in 12 out of 13 other domain test sets. On average, **our method outperforms AF by 10.02% and Full by 21.19%**. Encouragingly, **EDMRS outperforms Full by 106.37% in computer science and 51.89% in math**, reflecting that our method fully preserves important logical reasoning capabilities. The reason for AF's poor performance across both target and other domains is exactly as described in our paper: it selects "generalist" experts responsible for other domains rather than true "specialists" for the current domain, leading to severe catastrophic forgetting during training. We hope this addresses your concern.

---

> ### Author Response · Authors · 2025-11-28
> **Response to Reviewer 4Z84 (part2)**
>
> ### W2: Experiments include only one dataset
> As mentioned above, we addressed this by adding the new medical domain dataset, yielding significant results that strongly support our method.
>
> ### W3: Results compared to the baselines are within 1%
> As noted above, we addressed this by introducing the medical domain dataset and extending our strategy to all 48 layers. Our strategy achieves target domain performance comparable to Full (0.6034 vs. 0.6070, only 0.6% gap) and significantly outperforms AF (0.6034 vs. 0.4848, +24.46%), while **outperforming AF by 10.02% and Full by 21.19%** on MMLU-Pro. Encouragingly, **EDMRS surpasses Full by 106.37% in computer science and 51.89% in math**, demonstrating substantial preservation of critical logical reasoning capabilities.
>
> ### Q1: Why do the finetuning experiments only include a single layer?
> You are absolutely correct. As mentioned above, we applied our strategy to all layers in the new experiments, achieving highly significant results.
>
> ### Q2: The selected model is trained with 8 expert activated per token; any fine-tuning changing this strategy may give suboptimal results.
> As clarified above, we did not alter the 8-expert activation setting; it remains fully consistent with the model's pre-training configuration.
>
> ---
>
> We hope these responses fully address your concerns. We sincerely thank you again for your valuable feedback.
>
> [1] Pal A, Umapathi L K, Sankarasubbu M. MedMCQA: A large-scale multi-subject multi-choice dataset for medical domain question answering[C]//Conference on health, inference, and learning. PMLR, 2022: 248-260.

---

### Author Response · Authors · 2025-11-28
**Summary of Revisions and Key Improvements**

## **Summary of Revisions and Key Improvements**

We sincerely thank all reviewers for their insightful comments. We are encouraged by Reviewers 4Z84 and 5HY4's recognition of the **importance and advanced nature of our MoE interpretability research**. We also appreciate Reviewer RmuA's acknowledgment of our **clear theoretical framework with formal derivations** and our method's effectiveness in **maintaining task performance and enhancing cross-domain generalization**. In this revision, we have **updated the manuscript**, introduced a **new large-scale dataset**, extended experiments to **all layers**, and added comparisons with **PEFT methods like LoRA**, significantly enhancing our work's reliability and completeness.


**We have fully addressed ALL reviewers' concerns, especially the primary shared concern about marginal gains (≈1%)**. Our **new large-scale experiments demonstrate substantial improvements that far exceed the original results**: our method matches Full fine-tuning in the target domain while outperforming baselines by **+24.46% (vs. AF)** and **+21.19% (vs. Full)** on cross-domain tasks, with remarkable gains of **+106.37% in computer science** and **+51.89% in math**.


**Notably, beyond mitigating catastrophic forgetting**, our work advances fundamental understanding of the widely used yet underexplored MoE interpretability. We established a monosemanticity analysis framework and derived three key insights, offering vital theoretical perspectives and practical guidance for the community.

### New Dataset and Strong Experimental Results

We introduced **MedMCQA**, a large-scale medical dataset with 183k training samples [1]. For cross-domain evaluation, we tested on **13 distinct MMLU-Pro domains: biology, business, chemistry, computer science, economics, engineering, history, law, math, other, philosophy, physics, and psychology**.



| Method | biology | business | chemistry | computer science | economics | engineering | history | law | math | other | philosophy | physics | psychology | MMLU-Pro Avg |
| :--- | :--- | :--- | :--- | :--- | :--- | :--- | :--- | :--- | :--- | :--- | :--- | :--- | :--- | :--- |
| Full | 0.4686 | 0.3853 | 0.4364 | 0.2683 | 0.5284 | 0.3664 | 0.3911 | 0.2997 | 0.3494 | 0.3874 | 0.4148 | 0.4442 | 0.6529 | 0.4148 |
| AF | 0.6039 | 0.4284 | 0.4408 | 0.4951 | 0.5711 | 0.3736 | 0.4094 | 0.2852 | 0.4796 | 0.4773 | 0.3968 | 0.4742 | 0.5050 | 0.4570 |
| EDMRS | **0.6151** | **0.4715** | **0.4523** | **0.5537** | **0.6244** | **0.4118** | **0.5118** | **0.3143** | **0.5307** | **0.5011** | **0.4409** | **0.4788** | 0.6291 | **0.5027** |
| vs Full | **+31.26%** | **+22.37%** | **+3.64%** | **+106.37%** | **+18.17%** | **+12.39%** | **+30.86%** | **+4.87%** | **+51.89%** | **+29.35%** | **+6.29%** | **+7.79%** | -3.65% | **+21.19%** |
| vs AF | **+1.85%** | **+10.06%** | **+2.61%** | **+11.84%** | **+9.33%** | **+10.22%** | **+25.01%** | **+10.20%** | **+10.65%** | **+4.99%** | **+11.11%** | **+0.97%** | **+24.57%** | **+10.02%** |

Meanwhile, EDMRS performs extremely close to Full on the target domain (0.6034 vs. 0.6070, 0.6% gap) and significantly outperforms AF (+24.46%). **EDMRS achieves best results in 12 out of 13 other domain test sets, outperforming AF by 10.02% and Full by 21.19% on average**. Notably, **EDMRS outperforms Full by 106.37% in computer science and 51.89% in math**, demonstrating that EDMRS fully preserves important general logical reasoning capabilities. AF's poor performance results from selecting "generalist" experts responsible for other domains rather than "specialist" experts truly responsible for the current domain, causing severe catastrophic forgetting.

**Comparison with LoRA:**

| Method | MedMCQA | biology | business | chemistry | computer science | economics | engineering | history | law | math | other | philosophy | physics | psychology | MMLU-Pro Avg |
| :--- | :--- | :--- | :--- | :--- | :--- | :--- | :--- | :--- | :--- | :--- | :--- | :--- | :--- | :--- | :--- |
| LoRA | 0.5090 | 0.6109 | 0.4030 | 0.3993 | 0.4439 | 0.5983 | 0.3581 | 0.4514 | 0.2598 | 0.4123 | 0.4697 | 0.4369 | 0.4357 | 0.5927 | 0.4517 |
| EDMRS | **0.6034** | **0.6151** | **0.4715** | **0.4523** | **0.5537** | **0.6244** | **0.4118** | **0.5118** | **0.3143** | **0.5307** | **0.5011** | **0.4409** | **0.4788** | **0.6291** | **0.5027** |
| vs LoRA | **+18.55%** | **+0.69%** | **+17.00%** | **+13.27%** | **+24.74%** | **+4.36%** | **+15.00%** | **+13.38%** | **+20.98%** | **+28.12%** | **+6.69%** | **+0.92%** | **+9.89%** | **+6.14%** | **+11.30%** |

**EDMRS significantly outperforms the classic LoRA, achieving superior performance in the target domain (+18.55%) and across all 13 other domains (+11.30%)**, confirming our method's superiority.

---
[1] Pal A, Umapathi L K, Sankarasubbu M. MedMCQA: A large-scale multi-subject multi-choice dataset for medical domain question answering[C]//Conference on health, inference, and learning. PMLR, 2022: 248-260.

---

### Note · Authors · 2026-01-06

I have read and agree with the venue's withdrawal policy on behalf of myself and my co-authors.